# Fabrication, Optimization, and Performance of a TiO_2_ Coated Bentonite Membrane for Produced Water Treatment: Effect of Grafting Time

**DOI:** 10.3390/membranes11100739

**Published:** 2021-09-28

**Authors:** Mohamad Izrin Mohamad Esham, Abdul Latif Ahmad, Mohd Hafiz Dzarfan Othman

**Affiliations:** 1School of Chemical Engineering, Engineering Campus, Universiti Sains Malaysia (USM), Nibong Tebal 14300, Pulau Pinang, Malaysia; mohamadizrin@student.usm.my; 2Advanced Membrane Technology Research Centre (AMTEC), School of Chemical and Energy Engineering, Universiti Teknologi Malaysia (UTM), Johor Bahru 81310, Johor, Malaysia; hafiz@petroleum.utm.my

**Keywords:** bentonite membrane, titanium dioxide nanoparticle, produced water

## Abstract

The main problem usually faced by commercial ceramic membranes in the treatment of produced water (PW) is low water flux even though ceramic membrane was well-known with their excellent mechanical, thermal, and chemical properties. In the process of minimizing the problem faced by commercial ceramic membranes, titanium dioxide (TiO_2_) nanocomposites, which synthesized via a sol-gel method, were deposited on the active layer of the hydrolysed bentonite membrane. This paper studied the influence of grafting time of TiO_2_ nanocomposite on the properties and performance of the coated bentonite membranes. Several characterizations, which are Fourier transform infrared (FTIR), scanning electron microscopy (SEM), energy-dispersive X-ray Spectroscopy (EDX), contact angle, porosity, and average pore size, were applied to both pristine and coated bentonite membranes to compare the properties of the membranes. The deposition of TiO_2_ nanoparticles on the surface of the coated bentonite membranes was successfully confirmed by the characterization results. The pure water flux performance showed an increment from 262.29 L h^−1^ m^−^² bar^−1^ (pristine bentonite membrane) to 337.05 L h^−1^ m^−^² bar^−1^ (Ti-Ben 30) and 438.33 L h^−1^ m^−^² bar^−1^ (Ti-Ben 60) as the grafting time increase but when the grafting time reached 90 min (Ti-Ben 90), the pure water flux was decreased to 214.22 L h^−1^ m^−^² bar^−1^ which is lower than the pristine membrane. The oil rejection performance also revealed an increase in the oil rejection performance from 95 to 99%. These findings can be a good example to further studies and exploit the advantages of modified ceramic membranes in PW treatment.

## 1. Introduction

Produced water (PW), a troublesome by-product, can be defined as formation water located in the reservoirs below the hydrocarbon layer. Once the reservoirs become mature and experience water cuts, the reservoirs will produce PW during oil and gas extraction. PW contain various complex compounds. However, it can be grouped into inorganic and organic compounds such as dispersed and dissolved oils, grease, chemicals, salts, dissolved gases, anions, and cations [1]. PW also contain hazardous and radioactive heavy metals, such as chromium, lead, barium, uranium, and cadmium [2]. Other than that, some microorganisms may also live in the PW since PW has nutrients in it [3]. Hence, the treatment of the PW must meet the offshore and onshore discharge rules and regulations before being released into the environment. If not, PW may result in severe conflicting consequence on our flora and fauna in the marine and land environment [4,5]. What make more concern, when some of the previous studies have declared that PW have become one of the largest waste stream in oil and gas industry [6,7]. One of the previous studies reported that the volume of the PW is 10 times larger than the extracted hydrocarbon. The other study also reported that matured oil fields such as oil fields in Danish North Sea may be one of the causes of increase production of PW, other than the demand of oil and gas. Various standalone or combined physical, biological and chemical technique were proposed for the treatment of PW either by removing the suspended solids, de-oiling, removal of iron, desalting the organic compounds and softening the PW. The example of physical treatment are activated carbon [8] which used as adsorbent media to remove organic compounds like BTEX (Benzene, Toluene, Ethylbenzene and Xylene), hydrocyclone [9] for the separation of solids in PW based on their density, electrodialysis(ED) [10] to separate either anions or cations. Other than that, physical treatment also includes precipitation [11], ion-exchange [12], American Petroleum Institute “API” separator [13], deep bed filter [14], aeration [15], sedimentation [16] and also thermal desalination [17]. As for the chemical treatment, the example are coagulation and flocculation [18,19,20], and lastly biological treatment [21,22] which is superior in removing anaerobic and aerobic microorganisms.

In the beginning of the 20th century, membrane technology had become one of the most effective methods introduced in the oil and gas industry [23,24,25]. Membranes usually will be divided into two groups, which are inorganic and organic membranes, based on their starting materials. Polymeric membranes can be classified in the organic membranes group, while ceramic, metal oxides, and metallic membranes are inorganic membranes. Previous studies have shown some interest in utilizing the ceramic membranes in the PW treatment without pre-treatment process [26,27], since the ceramic membranes have great advantages in their mechanical, chemical and thermal characteristics. Furthermore, ceramic membranes also have the advantages in harsh environment, such as backwashing, autoclaving and multiple cleaning process due to their high chemical and thermal stability. The lifetime of the ceramic membrane is also longer and better than the polymeric membrane. In the production of ceramic membranes in the industry and in research studies, there are numerous techniques that can be applied. Flat sheet ceramic membranes are usually fabricated via the tape casting and pressing technique, while for the fabrication of hollow fiber ceramic membrane the technique that will be applied to produce it are the phase inversion and slip casting technique [28]. There are numerous studies which have focused on the application of ceramic hollow fiber membrane on the PW treatment. For example, there was a study using a low-cost alumina-spinel composite hollow fiber microfiltration membrane for the pretreatment of synthetic oily saline produced water [29]. The results illustrated that the prepared membrane efficiently remove the oil droplet in PW by 92.41%. The turbidity was also found to be reduced around 92%. Another study used a low cost superhydrophobic-superoleophilic kaolin-based hollow fiber ceramic membrane for the recovery of oil from synthetic produced water [30]. The results showed an average flux of 80 L m^−2^ h^−1^ and 90% of oil recovery. Meanwhile, a study conducted using mullite–kaolinite powder was used to treat PW from an oil refinery through which >97% TOC removal was reported [31].

In the past few years, the interest in fabrication of ceramic membranes had turned to the surface modification of the membrane, since the ceramic membranes itself have superior in most of the physical and chemical of properties. Hence, new studies were more focused in increasing the flux and rejection performance. A previous study successfully achieved better oil rejection and flux by implementing a graphene oxide (GO) nanoparticle, modifying the commercial ceramic alumina (Al_2_O_3_) membrane by the vacuum technique [32]. Next, kaolin hollow fiber membranes were dip coated with nano-Al_2_O_3_ for oil and water separation [33]. As expected, the coated membranes demonstrated an improvement in hydrophilicity, resulting in good oil removal efficiency up to 98%. A study also reported that polyamide/titanium dioxide (TiO_2_) composite hollow fiber nanofiltration (NF) membranes were successfully fabricated via the sol-gel method [34]. The coated NF membrane was prepared by immersing it vertically in the prepared TiO_2_ sol. The resulting coated NF membrane outperformed the pristine membranes by showing higher fluxes as high as 105.5 L m^−2^ h^−1^ at four bar and higher oil rejection greater than 95%. In the next study, an omniphobic mullite hollow fiber membrane with flower-like TiO_2_ was successfully fabricated, and the membrane surface wettability were improved [35]. In another study, α-Al_2_O_3_ membranes were coated with TiO_2_ using a simple dip-coating process with tetraethyl orthosilicate (TEOS) as a binder to increase the hydrophilic characteristic of the membrane surface [36]. After testing, the methylene blue (MetB) removal efficiency (up to 91%), as well as the flux of the modified membrane, exceeded the uncoated membrane. In this study, the authors were inspired by the sol dip-coating and sol-gel method reported in the literature and aimed to coat the bentonite membranes with TiO_2_ nanoparticles on the membranes’ surface via dip coating and heat for calcination. Hence, the aim of this study is to modify the surface of the bentonite membranes with nanoparticles via the combination of sol-gel dip coating and calcination techniques, as inspired by the method used in the reported previous study. The concept of sol gel dip-coating bentonite membranes using nanoparticles can be the pioneer of study regarding to the surface modification of bentonite membranes for the treatment of PW since there was not reported before. Thus, this study will be more focused on the effect of grafting time (30, 60, and 90 min) of TiO_2_ nanoparticles on bentonite membranes to find the optimum grafting time via the characterization and the performance of the coated membrane toward oil rejection and pure water flux, using synthetic PW.

## 2. Materials and Methods

### 2.1. Materials

The bentonite membranes with an average pore size of 1.75 µm were from previous study [37]. Titanium butoxide (Ti(OC_4_H_9_)_4_), absolute ethanol (EtOH, 99.5%, A.R.), Hydrochloric acid (HCl, 37%), glacial acetic acid, and distilled water (H_2_O) were used during this study without any further purification.

### 2.2. Preparation of TiO_2_ Sol

In the preparation of TiO_2_ sol solution, 10 mL of titanium butoxide and 35 mL of ethanol were mixed together using a magnetic stirrer for 10 min in room temperature to obtain the TiO_2_ precursor solution. Concurrently, in another magnetic stirrer, 2 mL of glacial acetic acid, 10 mL of distilled water, and another 35 mL of ethanol were stirred for 10 min, then few drops of hydrochloric acid were used to adjust the pH to pH = 3. After that, the prepared TiO_2_ precursor was slowly added to the second solution, then the stirring continued for another 1 h to obtain the TiO_2_ sol. The prepared TiO_2_ sol then was aged for three days at room temperature prior to the usage.

### 2.3. Deposition of TiO_2_ Nanoparticle on Bentonite Membrane

Before the process of dip-coating, the bentonite membranes were potted at both ends using polytetrafluoroethylene (PTFE) film tape to avoid the TiO_2_ sol solution from entering the lumen of the membranes. The potted membranes were dipped into ethanol for 30 min to enhance the OH bond on the surface of the membranes. The process of dip-coating into the TiO_2_ sol solution begun with various grafting times (30, 60, and 90 min). Then, the coated bentonite membranes were dried for 1 h at room temperature before the repeated cycle of dip-coating process. In this study, the cycle of dip-coating process was fixed to three times for each membrane after the drying process, as suggested in a previous study [38]. After that, the membranes were dried in an oven for 30 min to remove the moisture content. Lastly, the membranes were calcined with a fixed calcination temperature of 400 °C, as suggested by previous study which is around 400–450 °C [39,40,41], for 1 h to remove all of the organic chemicals and crystallize the anatase-TiO_2_. Table 1 shows the prepared coated bentonite membranes with different grafting times.

### 2.4. Membrane Characterization

Fourier transform infrared spectroscopy (FTIR, Thermo Scientific Nicolet Nexus 670, Waltham, MA, USA) in the range of 4000–500 cm^−1^ using the attenuated total reflectance technique (ATR) was used to check the functional groups related to TiO_2_ coated bentonite membranes. Moreover, scanning electron microscopy (SEM, HITACHI S-3000 N, Hitachi Ltd., Tokyo, Japan) was employed to observe membranes’ surfaces and cross-sections. Moreover, energy-dispersive X-ray Spectroscopy (EDX) also was utilized at 10 KeV to examine the TiO_2_ nanoparticle distribution and density on the coated membranes. Zeta Potential (Malvern Ver. 2.3, Malvern Instruments Ltd., 2008, Malvern, UK) were performed to study the charges of the TiO_2_ nanoparticle and oil particle, also the size of oil particle. The pore size of membranes was analyzed using Capillary Flow Porometer (Porolux 1000, Nazareth, Belgium) by following “dry up-wet up” method. The pressure of gas was gradually increased from 1 to 5 bar and the gas flow rate was recorded. Bubble point was recorded when the pressure was high enough to remove the liquid out from the largest pores. As pressure gradually increased, smaller pores become unblocked by liquid and the gas flow rate increased until the whole sample was completely dry. The cumulative pressure was used to calculate pore size distribution and average pore size. The pore sizes were estimated using perfluorether (porefil) solution. As for the porosity, the prepared pristine and coated membranes were immersed in the distilled water for 12 h and then weighed immediately after blotting. After that, the membranes were dried in oven for 12 h before dry weighed recorded. The porosity of the membranes was calculated by using Equation (1):(1)$$\varepsilon =\frac{\frac{m_{w}}{\rho_{dw}}}{\frac{m_{w}}{\rho_{dw}}+\frac{m_{d}}{\rho_{b}}}\times 100\%$$
where *ε* indicates the porosity of membrane (%), *m_d_* and *m_w_* are the mass of the dry and wetted membranes (g), respectively, and ρ*_b_* and ρ*_dw_* are the density of the bentonite and density of distilled water (g cm^−3^), respectively. The calculation of porosity was taken with five measurements and averaged to minimize error.

The contact angles of the membranes were estimated using the goniometer (Rame-Hart 250F-1, Succasunna, NJ, USA), to study the wettability of pristine and coated membranes. The device was used to perform a static contact angle test for the membranes. The contact angle test was run in two conditions, which are in air condition and underwater condition. Distilled water and oil droplet was used as the droplet dropped on the membranes’ surface for each contact angle test. The contact angle device was equipped with a camera, which recorded every image/video of the droplet, starting from before the water droplet drop to the membranes’ surface until there is no movement of the droplet. For each membrane, the droplets were dropped at 10 difference parts of the membrane surface. Then, an average value of contact angle was recorded and analyzed. Prior to the contact angle test for underwater conditions, the prepared pristine and coated bentonite membranes will be placed in a black box setup, as shown in Figure 1, for 1 h. A 36W UV lamp (Brand: Philips, Tokyo, Japan) is used as the light source and the illumination distance is 35 cm.

### 2.5. Water Flux Permeation and Oil Rejection Performance

A crossflow filtration system as illustrated in Figure 2 was used to perform a water flux and oil rejection performance of pristine bentonite and coated bentonite with a length of 10 cm at a pressure of 3 bar with constant flow rate of 400 mL min^−1^ and constant room temperature. The water flux (*J_W_*) was obtained and calculated by using Equation (2):(2)$$J_{W}=\frac{V_{w}}{A\times t\times P}$$
where *V_W_* is the permeate of water permeate through the membrane (L), *A* is the membrane surface area (m^2^), *t* is the sampling time (h), and *P* is the transmembrane pressure (bar).

The rejection (*R*) of the oil was measured under the same condition as the water permeation test to evaluate the benchmarking of oil rejection. The rejection rates were obtained and calculated according to Equation (3).
(3)$$R=\frac{\left(C_{f}-C_{p}\right)}{C_{f}}\times 100\%$$
where *R* is the rejection rate of oil (%), *C_f_* is the oil concentration of feed (mg/L) and *C_p_* is the oil concentration of permeate (mg/L). The concentration of oil in the produced water before and after ultrafiltration using UV-visible spectrophotometer (Thermo Scientific, Genesys 10S, Waltham, MA, USA) at a wavelength of 254 nm. The synthetic produced water was produced by mixing the heavy crude oil obtained from PETRONAS Melaka Refinery Complex and distilled water with the concentration of 1000 ppm. The concentration of surfactant (SDS) used in the mixing process was based on the ratio of oil and surfactant which is 9:1. The oil content was tested and analyzed prior to the oil rejection performance test.

## 3. Results and Discussion

### 3.1. Characterization of Pristine Bentonite and Coated Bentonite Membranes

#### 3.1.1. FTIR Analysis

Figure 3 shows the FTIR analysis for the pristine bentonite membranes and coated bentonite membranes (Ti-Ben 30, Ti-Ben 60 and Ti-Ben 90). It can be clearly seen that Ti-Ben 30, Ti-Ben 60 and Ti-Ben 90 had the peak of Ti-O at ~1000 cm^−1^ and peak of Ti–OH at ~1620 cm^−1^. Other than that, the peak at ~2950 which also can be seen on all of the TiO_2_ coated membrane are the characteristic of –CH_2_. All the peaks for coated bentonite membranes, however, was not witnessed on the pristine bentonite membrane FTIR spectra. This proves that the membrane itself has no peak at that wavenumber and that those peaks observed are due to the TiO_2_ deposition solely. Furthermore, by comparing the coated membranes, we can find that the peak of Ti–O, Ti–OH and –CH_2_ peaks increased as the grafting time increased. The results obtained are similar to the previous study. The peak intensity increased as the concentration of the TiO_2_ on the membrane increased [42].

#### 3.1.2. Surface and Cross-Sectional Morphology Analysis

The SEM images were used to investigate the morphology of the surfaces and cross-sections of the membranes. Figure 4 shows the SEM images of surfaces and the cross-section images of Ti-Ben 30, Ti-Ben 60, Ti-Ben 90 membranes. The surface SEM images were all taken at 2000× magnification whereas the cross-section images were taken at 400× and 2000× magnification. The white particles observed on the coated membranes surfaces were identified as TiO_2_ nanoparticles. This was deduced as they showed most on membrane Ti-Ben 90 and the least on membrane Ti-Ben 30.

The main observation which can be concluded from the surface SEM images is the different deposition densities of TiO_2_ on the membranes according to the grafting time. At the grafting time of 30 min, the surface SEM image of Ti-Ben 30 showed some parts of the membrane surface were started to be covered by the TiO_2_ nanoparticles. However, as the grafting time increased to 60 min (Ti-Ben 60) and 90 min (Ti-Ben 90), the TiO_2_ nanoparticles started to cover almost every part of the membrane surface. As we can see in Figure 4(c1), some of the TiO_2_ nanoparticles have entered the pore of the membrane which led to partial blockage on some of the membrane pores. It is notable that the TiO_2_ nanoparticles only covered some part of the coated bentonite membranes’ surface similar to the previous study [41]. As the study reported, the partially coated TiO_2_ would help in increasing the flux of the coated membrane. To support the SEM results of Ti-Ben 90 partial blockage, the cross sectional FESEM and EDX was analyzed, as shown in Figure 5. As can be seen in the Figure 5b, some of TiO_2_ was detected inside the pore of the coated membrane, as the light blue color is also shown on the membrane surface.

#### 3.1.3. Wettability Analysis

The wettability properties of the pristine and coated membranes were analyzed by measuring the water contact angle (WCA) and oil contact angle (OCA), under both air and underwater conditions. These analysis also were also identified to help in clarifying the changes in pure water flux and oil flux performance [41]. It should be noted here that the wettability test for WCA for air and underwater was only shown for Ti-Ben 30, since the results of images for Ti-Ben 60 and Ti-Ben 90 displayed the same results. As shown in Figure 6 and Figure 7, these contact angle values were measured at the first moment (1 s) the water droplets drop onto the membrane surface, since the water droplets were quickly dissolved into the membrane. In air condition, the WCA of Ti-Ben 30 was 10.21° ± 0.54° lower than pristine bentonite membrane which was 30.54° ± 0.17°. Then, after several seconds, the WCA of both pristine and Ti-Ben 30 turned to 0°. In underwater condition, the water droplets were quickly dissolved into the Ti-Ben 30 membrane at the first moment the water droplets touch the membrane surface, while there is still an angle on the pristine bentonite membrane which is 22.59° ± 0.71°.

Figure 8 displays the OCA of pristine and coated membranes for air and underwater conditions. Surprisingly, no static OCA can be measured for the pristine membrane for both conditions. As for coated membranes, the value of the OCA did not reach the oleophobic range when tested on air condition, which are 56.51° ± 0.27° (Ti-Ben 30), 84.13° ± 0.20° (Ti-Ben 60) and 90.30° ± 0.10° (Ti-Ben 90), but as for underwater condition with UV light illumination, the OCA values for all coated membranes increased drastically to 130.24° ± 0.10° (Ti-Ben 30), 147.00° ± 0.05° (Ti-Ben 60) and 155.50° ± 0.10°, as the grafting time increased.

These results showed great enhancement in the wettability of the coated bentonite membranes. The most interesting part is that the coated bentonite membranes tend to switch the wettability of membrane from hydrophilic membrane to underwater superhydrophilic and superoleophobic after being illuminated with UV light. A similar trend of findings was reported on photoinduced switchable underwater superoleophobicity-superoleophilicity on laser modified titanium surfaces [43]. Similarly, as the TiO_2_ surface radiated with UV while immersed in water, the water contact angle was found to be around 2° which also can be considered as super-hydrophilic surface. Another study also showed the membrane ability to switch the wettability to underwater superoleophobic as the UV light illuminated the TiO_2_/TCNC membrane [44].

This wettability enhancement can be explained by two phenomenon of the reactivity of TiO_2_ toward the UV light as shown in Figure 9. The first phenomenon, as the TiO_2_ surface being illuminate with UV light in the water, there will be the appearance of holes and electrons. The electrons that have been photogenerated will react with oxygen molecules in the surrounding, then produce superoxide radical anions, while the holes will be filled up by water molecules to produce OH radicals. These generated groups are very strong oxidizing agents which help in repelling the oil particles [45].

The other phenomenon is the formation of water film on the membrane surface. In this particular case, the electrons and holes were also generated. However, they will undergo different reaction. The photogenerated electrons tend to reduced Ti(IV) cations to Ti(III) and the holes will be oxidize into O_2_^−^ anions. During this mechanism, the oxygen will be thrown out and a group of oxygen vacancies are produced on the surface. Finally, the water molecules will occupy the empty sites and adsorbed OH radicals to create the water film, thus increase the hydrophilicity of the membrane [46,47].

### 3.2. Performance Tests of Pristine Bentonite Membrane and TiO_2_ Coated Bentonite Membranes

Figure 10 shows the pure water flux of pristine bentonite membranes and coated bentonite membranes (Ti-Ben 30, Ti-Ben 60 and Ti-Ben 90) while Table 2 shows the average pore size, porosity and coating thickness of the prepared membranes. As shown in the figure, two observations can be seen, where the flux decreased with time for all membranes and the changes of flux as the grafting time was increased. In the beginning of the filtration, Ti-Ben 30 and Ti-Ben 60 have the highest pure water flux of about ~2700 L h^−1^ m^−^² bar^−1^ followed by pristine bentonite membranes and Ti-Ben 90, which are about ~1800 and ~1300 L h^−1^ m^−^² bar^−1^, respectively. However, as the time increased, different steady-state flux time were obtained for each membrane, for which the fastest time are pristine bentonite membrane (*t* = 50 min), followed by Ti-Ben 90 and Ti-Ben 30 (*t* = 70 min), and lastly Ti-Ben 60 (*t* = 100 min). It should be noted here that the pure water flux and oil rejection performance were tested starting on the first minute of the filtration without the flux stabilization process. This is why the fluxes were decreased as the time increased, so that we can observed the steady-state flux time for each tested membrane. Results also showed that the pure water flux at the end of the filtration for Ti-Ben 30 and Ti-Ben 60 were increased when compared with the pristine membranes, which were around 28% (337.05 L h^−1^ m^−^² bar^−1^) and around 67% (438.33 L h^−1^ m^−^² bar^−1^), respectively. However, the increment trend of pure water flux did not continue as the grafting time increased to 90 min (Ti-Ben 90) when the value decreased around 214.22 L h^−1^ m^−^² bar^−^^1^, which was also even lower than the pristine bentonite membranes. When comparing the Ti-Ben 30 and Ti-Ben 60, the pure water results showed Ti-Ben 60 was performed more effectively than Ti-Ben 30. This can be explained by the coating thickness of TiO_2_ nanoparticles on the membrane’s surface. Clearly, we can see from SEM images that the coating thickness of TiO_2_ on the Ti-Ben 60 membranes’ surface much better in quantity than Ti-Ben 30 (Figure 4). It was found that the increase of the coating thickness between the Ti-Ben 30 and Ti-Ben 60 is very marginal (10 and 12 µm). Thus, it is expected that the diffusion resistance between these two membranes is very little. However, Ti-Ben 60 would have higher concentration of TiO_2_ which contributed to the higher pure water flux. The sufficient coating thickness of TiO_2_ nanoparticles on the membranes’ surface help in better acceptance of water droplet passing through the coated membranes [48]. In contrast, there are two explanations that can be derived from the declination of pure water flux for Ti-Ben 90. The first explanation is that the membranes’ pore blockage resulting from the high concentration of TiO_2_ nanoparticles on the membranes’ surface. As compared to other coated membranes, Ti-Ben 90 had the highest coating thickness of TiO_2_ nanoparticles on the membranes’ surface. The coating thickness is almost double (23 µm), which contribute to the higher diffusion resistance. As shown in Table 2, even though the thickness of TiO_2_ coating increased the hydrophilicity of the membrane. However, the average pore size (0.63 µm) and porosity (45%) was expected to be drastically decline due to the attachment of nanoparticle of the membranes’ pores. The SEM and FESEM results on the cross-section (Figure 4(c1) and Figure 5b) also showed that some of the TiO_2_ nanoparticle had entered the membrane pores which resulting of the pore blockage. It can be concluded that even though the coated membranes have higher hydrophilicity than the pristine membranes, they would not give a higher advantage in flux to the coated membranes if there were a drastically decreased in membrane porosity. These results are also similar to the previous study, wherein they reported that the flux of the coated membrane did not only affect by the membrane hydrophilicity and they argue that even though the hydrophilicity increased, the improved membranes can be in a disadvantage situation if the membranes’ pores were blocked by the coated hydrophilic nanoparticles [49]. The second explanation that also affecting the declination of the flux for Ti-Ben 90 is the interaction force between the hydrophilic coated membrane and the feed water. This interaction force restricting the flow of the feed water to pass through the membranes’ surface [50].

The oil rejection test was conducted to evaluate the oil removal capacity of the membranes. Figure 11 shows the permeate flux and oil removal efficiency obtained for the pristine bentonite membranes and coated bentonite membranes (Ti-Ben 30, Ti-Ben 60 and Ti-Ben 90). The permeate flux and oil rejection through pristine bentonite membranes was 183.15 h^−1^ m^−^² bar^−1^ and 95%, respectively. The high oil rejection in the pristine bentonite membrane was further enhanced in all the coated membranes reporting 97, 98.5 and 99% for Ti-Ben 30, Ti-Ben 60 and Ti-Ben 90; respectively. As expected, the oil rejection efficiencies were increased as grafting time of TiO_2_ on the membrane surface increased.

This can be explained by several points. The first point is the pore size of the coated membranes, whereas the oil particle size (1 µm) is more than the pore size of the coated membranes (0.95, 0.89 and 0.63 µm), as shown in Table 2 and Table 3. Thus, this will give an advantage for the coated membrane surface to not allow the oil particle to enter the membrane. Secondly, the zeta potential of the TiO_2_ particles was measured between −33 to −37.8 mV as shown in Figure 12. Since TiO_2_ nanoparticles is a negatively-charge particles, hence TiO_2_ nanoparticles on the membranes’ surface plays a role in repelling the negatively-charged oil droplets which measured around −5 to −6 mV as shown in Table 3, which also resulting in preventing the oil particle to enter the pores of the membranes [51], thereby increasing the oil rejection performance.

Figure 13 shows the long-term permeate flux evaluation of the TiO_2_ coated bentonite membrane with optimum grafting time which is Ti-Ben 30. It can be seen that Ti-Ben 30 able to maintain the flux around 300 L h^−1^ m^−^² bar^−1^ with good rejection between 98–99% of oil as the permeate sample were analyzed every 60 min, as shown in Table 4. This indicate that TiO_2_ coated membrane with optimum grafting time have the potential for a long-term PW treatment. Table 5 shows the properties of Ti-Ben 30 before filtration, in first 3 h and at the end of filtration. The result showed the differences results of OCA and zeta potential, which indicate that the TiO_2_ nanoparticle might leaching out a bit during the long-term filtration. This particular study about the long-term evaluation can be studied and discussed in the future on the durability of the TiO_2_ nanoparticle of the membrane surface.

## 4. Conclusions

TiO_2_ coated bentonite membranes were successfully prepared via the sol-gel dip coating technique using TiO_2_ nanoparticles. The technique of the deposition of TiO_2_ nanoparticles was confirmed via the characterization results. With the increasing of grafting time of TiO_2_ nanoparticles, the concentration of TiO_2_ nanoparticles on the membrane’s surface were increased as results shown in SEM results. However, when the grafting time increased to 90 min, the TiO_2_ started to disturb the membranes’ pores. The results also reported that the TiO_2_ distribution on the membranes’ surface was not fully-coated, however, it did not affect the performance of the membrane since the results displayed higher oil rejection and pure water flux than the pristine bentonite membranes. The Ti-Ben 60 with grafting time of 60 min displayed an outstanding pure water flux performance with 67% increment from the pristine bentonite membrane. Ti-Ben 60 also was selected as the best performance compared to pristine and other TiO_2_ coated membranes as it had higher oil rejection performance of 98.5%. Meanwhile, as for the contact angle results, the deposition of TiO_2_ nanoparticles on the membranes’ surface allowing the TiO_2_ coated bentonite membranes to switch the wettability from hydrophilic to underwater superhydrophilic and superoleophobic properties in underwater condition. It could be concluded that the surface coating of bentonite membranes with TiO_2_ nanoparticles has great potential for the removal of oil from industrial wastewaters, especially PW.

## Figures and Tables

**Figure 1 membranes-11-00739-f001:**
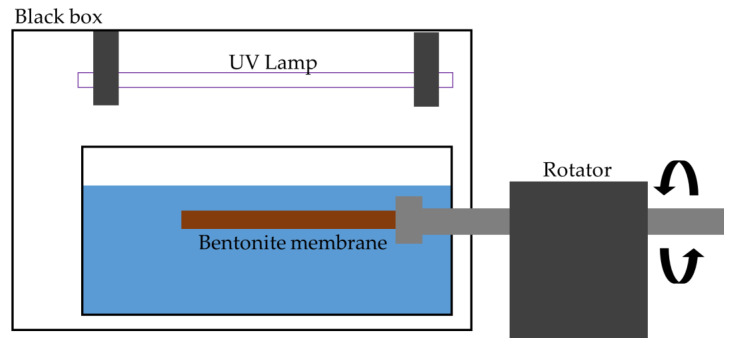
Black box setup for illumination of UV light.

**Figure 2 membranes-11-00739-f002:**
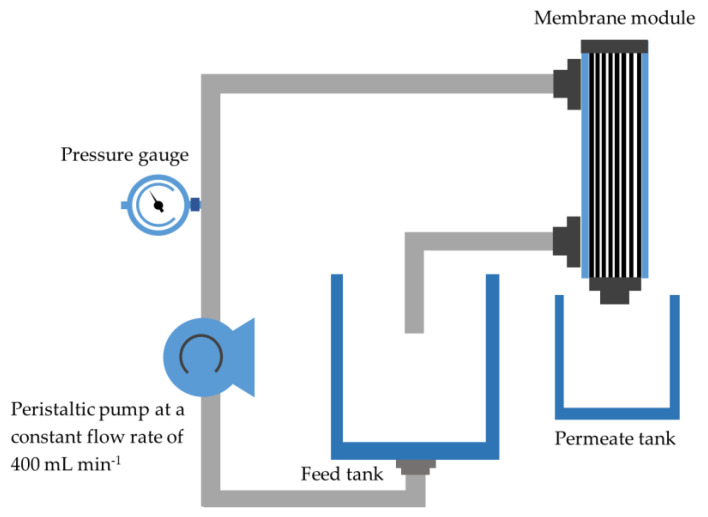
Crossflow filtration setup for produced water treatment.

**Figure 3 membranes-11-00739-f003:**
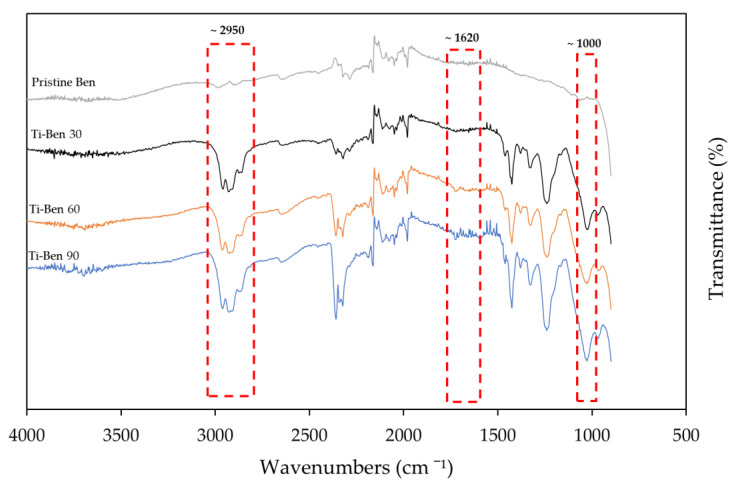
FTIR spectra for pristine bentonite membrane, Ti-Ben 30, Ti-Ben 60, and Ti-Ben 90.

**Figure 4 membranes-11-00739-f004:**
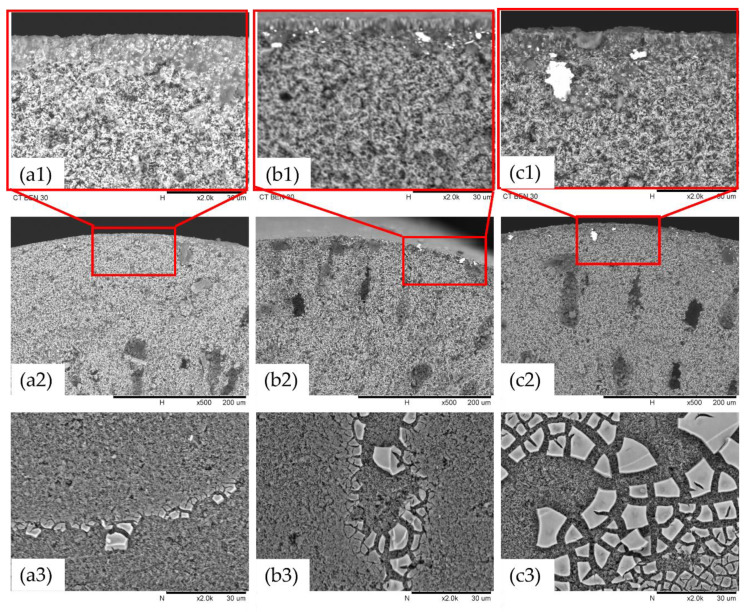
SEM images of (**a**) Ti-Ben 30, (**b**) Ti-Ben 60, and (**c**) Ti-Ben 90 on (1 and 2) cross-section of the membranes at 2000× and 500× magnification, respectively, and (3) surface of the membrane at 2000× magnification.

**Figure 5 membranes-11-00739-f005:**
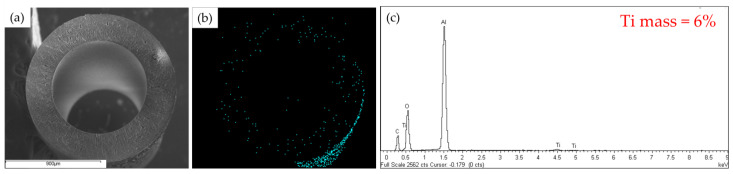
FESEM at magnification of 200× and Ti mapping images of Ti-Ben90, (**a**) cross sectional FESEM, (**b**) EDX Mapping of Ti, and (**c**) EDX result of Ti percentage.

**Figure 6 membranes-11-00739-f006:**
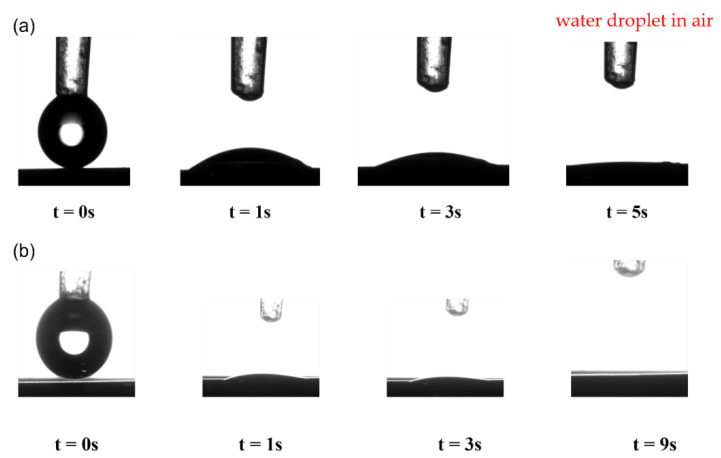
Images of water droplet on, (**a**) pristine bentonite membrane and (**b**) Ti-Ben 30, for air condition.

**Figure 7 membranes-11-00739-f007:**
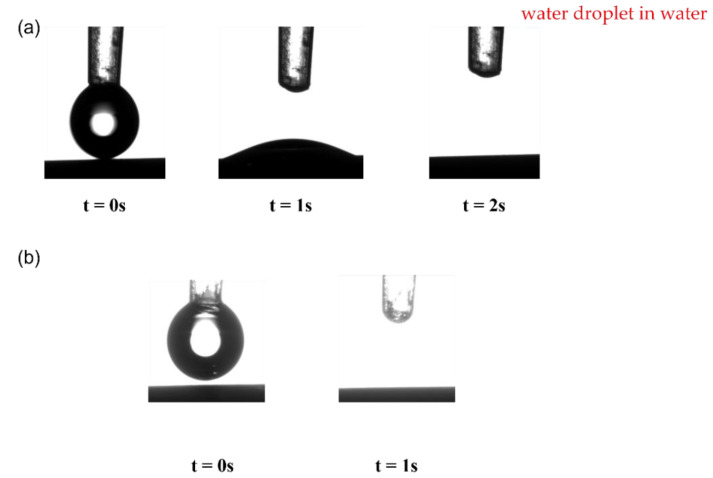
Images of water droplet on, (**a**) pristine bentonite membrane, and (**b**) Ti-Ben 30, for underwater condition.

**Figure 8 membranes-11-00739-f008:**
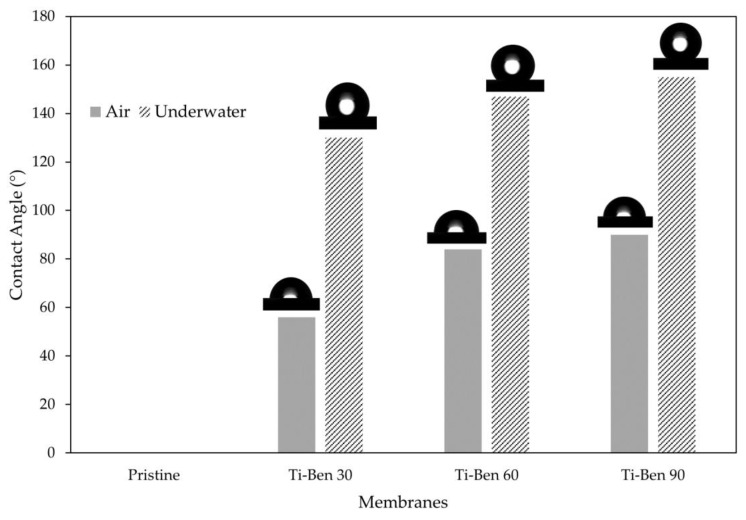
Oil contact angle values for each membrane for air and underwater condition.

**Figure 9 membranes-11-00739-f009:**
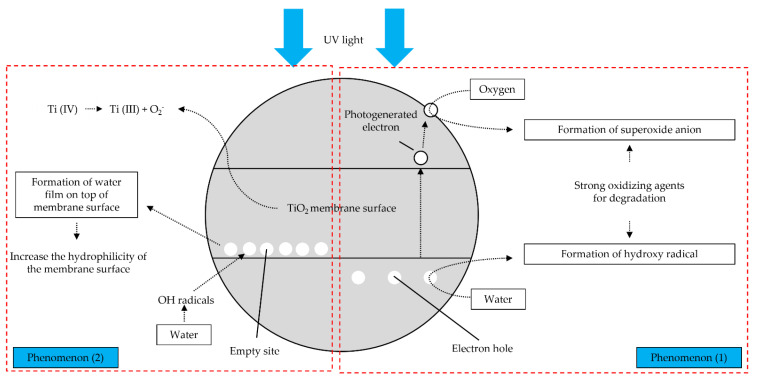
Photocatalytic (phenomenon 1) and hydrophilicity (phenomenon 2) mechanism.

**Figure 10 membranes-11-00739-f010:**
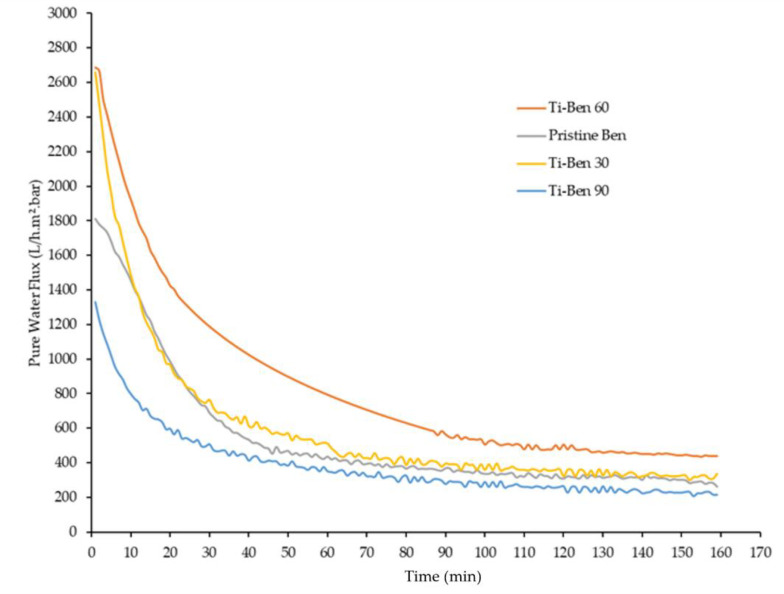
Pure water flux performance of pristine bentonite membrane, Ti-Ben 30, Ti-Ben 60, and Ti-Ben 90.

**Figure 11 membranes-11-00739-f011:**
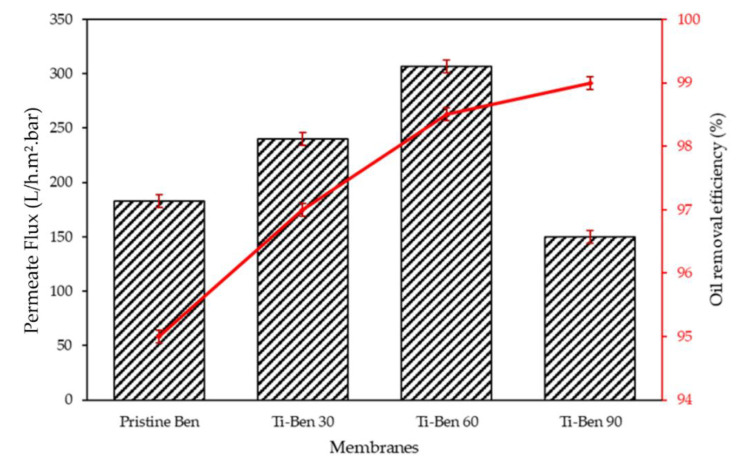
Permeate flux and oil removal performance of pristine bentonite membrane, Ti-Ben 30, Ti-Ben 60, and Ti-Ben 90.

**Figure 12 membranes-11-00739-f012:**
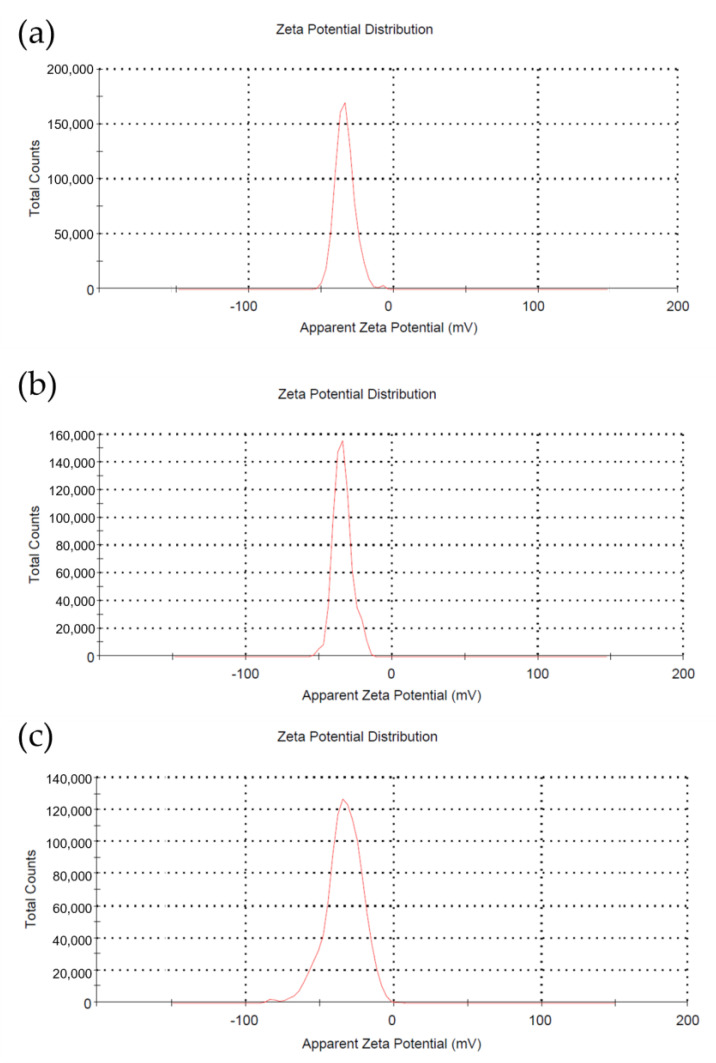
Zeta potential distribution of (**a**)Ti-Ben 30, (**b**) Ti-Ben 60, and (**c**) Ti-Ben 90.

**Figure 13 membranes-11-00739-f013:**
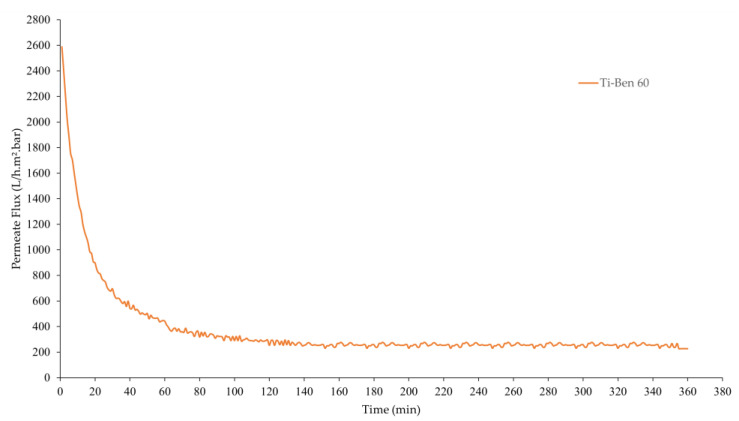
Permeate flux of Ti-Ben 60 for long-term evaluation.

**Table 1 membranes-11-00739-t001:** Coated bentonite membranes with different grafting time.

Membrane	Grafting Time (min)	Grafting Cycle (times)	Calcination Temperature (°C)
Pristine Ben	0	3	400
Ti-Ben 30	30	3	400
Ti-Ben 60	60	3	400
Ti-Ben 90	90	3	400

**Table 2 membranes-11-00739-t002:** Average pore size, porosity and coating thickness of the prepared membranes.

Membrane	Average Pore Size (µm)	Porosity (%)	Coating Thickness (µm)
Pristine Ben	1.75	62	-
Ti-Ben 30	0.92	61	10
Ti-Ben 60	0.89	58	12
Ti-Ben 90	0.63	45	23

**Table 3 membranes-11-00739-t003:** Properties of the synthetic PW.

Parameter	Value	Unit
Size of oil droplets	1.0	µm
pH	6	-
Zeta potential	−5 to −6	mV

**Table 4 membranes-11-00739-t004:** Permeate sample taken for every 60 min.

Time (min)	Rejection of Oil (%)
60	98
120	98
180	98.5
240	98.5
300	99
360	99

**Table 5 membranes-11-00739-t005:** Properties of Ti-Ben 30 in first 3 h and at the of filtration.

Properties	Before Filtration	In First 3 h	At the End of Filtration
OCA	147°	147°	130°
Zeta potential	−33 mV	−33 mV	−30 mV

## Data Availability

Not applicable.

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
