# Peer review of "Fabrication, Optimization, and Performance of a TiO2 Coated Bentonite Membrane for Produced Water Treatment: Effect of Grafting Time"

_membranes, 2021, doi:10.3390/membranes11100739_

Round 1
Reviewer 1 Report
The authors demonstrated the effect of grafting time in preparing TiO2 coated ceramic membrane for oil removal from produced water. The experimental work is not really much as other parameters of grafting are not studied to correlate the process. Meanwhile, some important characterizations are lacking to support the explanation made by the authors. Hence, I would suggest a major correction and the manuscript should be only accepted after the following comments are addressed.
Lines 31-51: Too many general descriptions on PW are included. Please summarize it and remove the repeating content.
Lines 38-40: The sentence is leading. Please rephrase it.
Lines 94-98: The reported work is not related to surface modification. Please remove it.
Line 135: Please measure and state the pH of the solution as it is unclear by just stating 2 drops of HCl is needed.
Section 2.3: As the authors only studied the grafting time, please justify the use of other parameters, including coating cycles and drying time and temperature. If those parameters are optimized in other works, please provide the citation.
Figure 1: Please clarify what BHFM means.
Section 2.5: As a crossflow filtration system is used, it is crucial to state the crossflow velocity used in testing. Please state it clearly for reference.
Figure 2: The setup is incomplete as it is lack of a flow meter to measure the flow rate.
Figure 3: Please redraw the diagram as it is unclear. Why did the authors not include the ATR-FTIR data for another two TiO2 coated membranes? It can be used to detect the intensity of TiO2 coated on membrane surface.
Line 214: The peak should be at particular “wavenumber”, not “frequency”. Please correct it.
Lines 235-237, Figure a1-d1: It is hardly to observe the TiO2 at the membrane cross-sectional as what mentioned by authors. Please include the EDX by detecting Ti element to visualize the location of TiO2 deposited on membrane surface and inside the pores.
Section 3.1.3: Even without TiO2, the high porosity of ceramic membrane causes difficulty in measuring water contact angle. Hence, the authors should provide oil contact angle to better understand the membrane hydrophilicity.
Lines 291-293: The authors should measure the membrane porosity using porometer or BET to support this statement.
Lines 293-294: Please provide the direct evidence to show the higher TiO2 concentration on membrane surface. Perhaps EDX mapping can help in this.
Lines 298-299: The authors should visualize the pore blockage in a clearer manner. EDX mapping should be included as suggested previously.
Figure 7: The unit of time is wrong. Why the PWF is decreasing with time as the ceramic membranes is strong and would not experience compression issue. Please explain this observation.
Lines 321-322: Please measure the sizes of oil particles and membrane pore sizes to support the statement. The authors should include the MWCOs of the membranes to classify the membrane and also help in explaining the rejection.
Lines 324-327: The authors need to measure the zeta potential of oil-in-water emulsion and also membrane surface to support the explanation. A simple assumption is a technical paper is just insufficient.
Section 3.2: It is doubted that the TiO2 coated membranes have high stability in a long-term filtration. Is the TiO2 physically or chemically bonded with membrane surface? The authors should run the testing for a long period and collect the water sample in feed tank to detect the amount of TiO2 leaking.
Reviewer 2 Report
The manuscript titled “Fabrication, optimization, and performance of a TiO2 coated bentonite membrane for roduced water treatment: Effect of grafting time (Mohamad Izrin Mohamad Esham, Abdul Latif Ahmad and Mohd Hafiz Dzarfan Othman)” is very interesting and attractive to many researchers. The manuscript is clear and well structured. The research problem is well explained and supported by appropriate experiments. I recommend this manuscript for publication.
Reviewer 3 Report
This paper studied the influence of grafting time of TiO2 nanocomposite on the properties and performance of the coated bentonite membranes. The pure water flux and oil rejection performance showed an increment as the grafting time increase. In general, this paper makes thoroughly study on bentonite membranes coated with TiO2 nanoparticles for treatment of produced water. I recommend to accept the manuscript after minor revision. A few things need to be addressed as shown below.
- The authors use UV-light to modify the membrane, but the mechanism of working principle of this experiment was not included in the manuscript. The reviewer will suggest to give more description.
- In section 3-2, performance test, Ti-Ben 60 showed higher water flux than Ti-Ben30, and the authors explained that was due to higher TiO2 concentration on the surface of Ti-Ben 60. However, thicker TiO2 also means greater diffusion resistance to water molecules. From this perspective, is it possible that the water flux decrease due to enhanced film thickness of TiO2 The reviewer will suggest the author to measure the film thickness of TiO2 layer to evaluate the diffusion resistance.
- In page 10, line 320, “To begin with, the pore size of the membranes, whereas the oil particle size is more than the pore size of the membranes”Please give the reference to the oil particle size.
- In Figure 4, the authors mentioned that membranes’ pores were blocked by the coated hydrophilic nanoparticles (Ti-Ben 90). Please mark the hole of the membrane and nanoparticles on the SEM images respectively.
- In Figure 8, the oil flux was increased as grafting time of TiO2 on the membrane surface increased. The oil should be hydrophobic, how comes that the flux was also increased?
Round 2
Reviewer 1 Report
The authors have addressed the comment accordingly and the manuscript is ready for publication.